# COVID-19 Testing Strategies for K-12 Schools in California: A Cost-Effectiveness Analysis

**DOI:** 10.3390/ijerph19159371

**Published:** 2022-07-30

**Authors:** Sigal Maya, Ryan McCorvie, Kathleen Jacobson, Priya B. Shete, Naomi Bardach, James G. Kahn

**Affiliations:** 1Philip R. Lee Institute for Health Policy Studies, University of California San Francisco, 490 Illinois St., Floor 7, San Francisco, CA 94158, USA; naomi.bardach@chhs.ca.gov (N.B.); jgkahn@ucsf.edu (J.G.K.); 2California Department of Public Health, Fresno, CA 95899, USA; ryan.mccorvie@cdph.ca.gov (R.M.); kathleen.jacobson@cdph.ca.gov (K.J.); 3Department of Medicine, University of California San Francisco, San Francisco, CA 94143, USA; priya.shete@ucsf.edu; 4Safe Schools for All, California Health and Human Services, Sacramento, CA 95814, USA

**Keywords:** COVID-19, screening, cost effectiveness

## Abstract

Public health officials must provide guidance on operating schools safely during the COVID-19 pandemic. Using data from April–December 2021, we conducted a cost-effectiveness analysis to assess six screening strategies for schools using SARS-CoV-2 antigen and PCR tests and varying screening frequencies for 1000 individuals. We estimated secondary infections averted, quality-adjusted life years (QALYs), cost per QALY gained, and unnecessary school days missed per infection averted. We conducted sensitivity analyses for the more transmissible Omicron variant. Weekly antigen testing with PCR follow-up for positives was the most cost-effective option given moderate transmission, adding 0.035 QALYs at a cost of USD 320,000 per QALY gained in the base case (R_eff_ = 1.1, prevalence = 0.2%). This strategy had the fewest needlessly missed school days (ten) per secondary infection averted. During widespread community transmission with Omicron (R_eff_ = 1.5, prevalence = 5.8%), twice weekly antigen testing with PCR follow-up led to 2.02 QALYs gained compared to no test and cost the least (USD 187,300), with 0.5 needlessly missed schooldays per infection averted. In periods of moderate community transmission, weekly antigen testing with PCR follow up can help reduce transmission in schools with minimal unnecessary days of school missed. During widespread community transmission, twice weekly antigen screening with PCR confirmation is the most cost-effective and efficient strategy. Schools may benefit from resources to implement routine asymptomatic testing during surges; benefits decline as community transmission declines.

## 1. Introduction

Coronavirus disease 2019 (COVID-19) was declared a global pandemic by the World Health Organization on 11 March 2020 [1]. Cases and deaths increased rapidly, both globally and in the United States [2,3]. In California, schools were closed for the remainder of the school year on 1 April 2020 [4], to help reduce the spread of SARS-CoV-2, the virus that causes COVID-19. It is estimated that this closure led to over 1.3 billion missed school days for over 24 million children of school age in the 2019–2020 school year alone [5]. Many schools struggled to re-open in 2020–2021, leading to even more learning days lost.

Schools are essential for children’s education and have many other benefits [5,6,7]. School attendance contributes to children’s social development and well-being [8] and allows access to resources such as food for those experiencing food insecurity [8,9]. Hence, re-opening and safely operating schools during the pandemic was an important step in mitigating adverse outcomes associated with extended closures.

In order to safely re-open schools, the state of California took a number of steps to prevent in-school transmission of SARS-CoV-2 [10] using a layered approach, including a robust program to support regular asymptomatic testing implementation [10,11]. While the use of asymptomatic screening has been widely regarded as an essential component of COVID-19 control, the effectiveness and cost-effectiveness of various testing strategies in schools remains unknown. Previous research in college campuses has shown that weekly screening can be an effective strategy to reduce transmission, with an estimated 39% reduction in the number of infections given a 90% sensitive test [12]. Utilizing highly sensitive polymerase chain reaction (PCR) tests was infeasible for most K-12 schools due to their long turnaround time and high cost [13]. Thus, many schools have increasingly been using rapid antigen tests, which allow them to offer a safer environment to resume in-person instruction [14]. However, the lack of universal guidance, resistance from families and staff to testing, and varying levels of resources led to high variability in the scope, implementation, and requirements of screening programs across schools [14].

Notably, the sensitivity of antigen tests is lower than that of PCR [15]; however, they provide results in approximately 15 min and can be performed by non-medical personnel, making them actionable in real-time [16]. Antigen tests make it possible to quickly identify and isolate individuals with high viral loads who are likely transmitting infection [17,18]. Increasing the frequency of testing may identify more infected individuals and reduce transmission with the crucial caveat that more frequent false positive test results will occur, leading to greater unnecessarily missed days of school for children.

In this study, we assessed two distinct tradeoffs related to school-based COVID-19 screening under various levels of community transmission: the cost-effectiveness (cost per quality-adjusted life year (QALY) gained) and the efficiency (days of school needlessly missed per secondary infection averted). We evaluated these outcomes for various screening approaches to help inform decision-makers in California and nationally as they work to provide safe in-person instruction during the COVID-19 pandemic.

## 2. Materials and Methods

### 2.1. Overview

We used a customized decision tree model to assess cost-effectiveness and efficiency of various COVID-19 screening strategies for K-12 schools to inform decision-making around school re-openings in California. Specifically, we aimed to evaluate the effect of screening frequency on outcomes. The following six testing strategies with different combinations of SARS-CoV-2 antigen and PCR tests and varying frequency of screening were assessed: (1) no testing; (2) once weekly antigen testing, henceforth 1×Ag; (3) twice weekly antigen testing, 2×Ag; (4) once weekly PCR testing, 1×PCR; (5) once weekly antigen testing with confirmatory PCR for positive antigen tests, 1×Ag/PCR; and (6) twice weekly antigen testing with confirmatory PCR for positive antigen tests, 2×Ag/PCR. For each strategy, we calculated net costs, number of secondary infections in school, QALYs lost, and number of unnecessarily missed school days. Efficiency was defined as the number of unnecessary days missed per secondary infection averted.

### 2.2. Decision Tree Design

Our model incorporates three possible health states for the population: susceptible, infected, or recovered (including vaccinated). For each of the six testing strategies, we modeled the probability of receiving a positive or negative test result based on test sensitivity and specificity. Those with positive tests began isolation for 10 days, while those with negative tests remained in school and continued to follow standard COVID-19 precautions such as wearing masks and socially distancing. If a confirmatory PCR test was conducted, we modeled one missed school day while awaiting a test result. These actions led to varying levels of transmission. We calculated the number of secondary infections in school, quality-adjusted life years (QALYs) lost, net costs incurred, and school days needlessly missed due to false positive tests with each strategy. The model was implemented in Excel (Microsoft 365) and probabilistic sensitivity analyses were conducted using @RISK v8 (Palisade Corporation, Raleigh, NC, USA). A simplified visualization of the model is presented in the Appendix A.

### 2.3. Key Assumptions and Model Inputs

We made several assumptions in constructing the model: (1) full 5-day schooling with continued infection control measures in place, including daily symptom screening; (2) base case epidemiologic parameters including the effective reproductive number (R_eff_) and duration of infectivity (ten days) based on data from April 2021; and (3), those who have been vaccinated or previously infected have a lower probability of becoming reinfected. These assumptions were evaluated in sensitivity analyses that included parameters based on more recent surges. We also assumed that index cases were equally likely to acquire SARS-CoV-2 on any given day. Once weekly tests were done on Mondays. Twice weekly tests were done on Mondays and Thursdays. The effect of different testing days on screening effectiveness are discussed.

Parameter values (Table 1) were identified through literature review, as well as stakeholder and expert inputs, and were assigned wide ranges for sensitivity analyses when necessary. While the base-case model inputs reflected moderate transmission periods, a sensitivity analysis was conducted using prevalence and transmissibility observed during the Omicron surge.

### 2.4. Epidemiologic and Health Inputs

*Prevalence of infected*: The proportion of school population with active infection at a given time was 0.2% in base case. This was conservatively based on the California daily case rate as of 13 April 2021 [19] (four new cases per 100 thousand people per day, an all-time low since the start of the pandemic in California), adjusted for a 25% symptomatic rate and 10-day infection duration.

*Prevalence of recovered*: The proportion of Californians who have either been previously infected or are fully vaccinated, measured as the 4-week average seroprevalence among adults as of 16 April 2021 [20]. This was 47% in base case and was varied in sensitivity analyses.

*Secondary transmission rate (R_eff_)*: The effective reproduction number in California as of 13 April 2021 [21], 1.1, was used in base case. This value indirectly accounts for the presence of infection prevention measures such as mask-wearing and social distancing. We assumed that eighty-one percent of these infections occurred within the school, with a student to teacher (and staff) ratio of 9:1, based on estimates of Bilinski et al. [22].

*QALYs lost per infection*: QALYs are a standardized measure incorporating mortality and morbidity due to a health condition. For COVID-19, we used an average of 0.078 QALYs lost per infection based on anticipated morbidity and mortality (discounted at 3%) [23,24].

### 2.5. Test Performance Inputs

*Antigen tests*: These are lateral flow assays, similar to at-home pregnancy tests. Saliva or nasal swab samples are placed on the card, and results can be read in approximately 15 min [16]. Among asymptomatic individuals, the sensitivity and specificity of the test is estimated as 58% and 99%, respectively, for all levels of viral load [25]. 

*Viral tests*: PCR tests detect viral genetic material in saliva or nasopharyngeal samples. We used an average sensitivity of 71% for asymptomatic screening when PCR was used as the only screening test [26]. This value accounts for the inability of PCR to detect infected individuals in the first few days of their infection, when their viral load is too low to be detected. When PCR tests were used to confirm a true positive antigen test, we assumed near-perfect performance with 99% sensitivity. Specificity was 99% in both cases [27].

### 2.6. Cost Inputs

*Cost of tests*: Rapid antigen test kits cost USD 5. PCR tests cost USD 21 to California schools, including the cost of testing kits and conducting the test in a laboratory (based on stakeholder input—the California Health and Human Services agency).

*Operational costs*: The cost of operating the testing program was approximately USD 4.5 per person per week for a single test, and for an average school population of 660 people [22] (including personnel time, based on stakeholder input). For twice weekly testing strategies, we increased this cost by a factor of 1.5 (to USD 6.8 per person per week) assuming some efficiencies with scale-up.

*Productivity losses*: These occurred when individuals were isolated due to a positive test and when secondary infections developed severe disease and could not attend school/work. We valued one missed day of a teacher or staff member at USD 320 [28], and one missed day of schooling for the students at USD 100 (assumed; +/−50% in sensitivity analyses). If isolated due to a positive test, the assumed productivity losses were 50% and 80%, respectively. If absent from school due to being severely ill, full productivity loss occurred.

*Medical costs*: We used a weighted average of acute COVID-19 treatment costs for different disease severities in adults [29], from asymptomatic to requiring intensive care, which was USD 3312 per infection (Appendix A). While children are less likely than adults to be hospitalized (which leads to the greatest medical costs), there is growing evidence that considerable proportions of children who contract SARS-CoV-2 can experience sustained symptoms following acute COVID-19 (long COVID) [30,31]. Therefore, our medical cost estimate is reasonable for a school population; this value was varied in sensitivity analyses.

### 2.7. Model Outputs

*Number of secondary infections at school*: Calculated per person tested for each strategy using the effective reproduction number, adjusted for screening frequency. We then calculated total QALYs lost with each screening strategy.

*Net cost to society*: This included cost of tests, operational costs of running a screening program, medical costs, and productivity losses. A school perspective was also explored which included testing and operational costs and loss of teacher productivity, but not medical costs or productivity losses associated with children’s missed schooldays.

*Total number of days missed*: Days of school or work missed due to being in isolation. A portion of this was *unnecessary days of school missed*, which occurred when non-infected individuals were required to isolate due to false positive tests. Additionally, we calculated the total number of infectious days at school for each testing strategy, which was individuals with false negative tests remaining in school.

We investigated two tradeoffs of costs versus benefits that are relevant to decision-making around schools. The first was cost-effectiveness, measured in net costs per QALY gained (incremental cost-effectiveness ratio (ICER)) when applicable. Second, we assessed the burden of false positive tests by calculating unnecessarily missed school days per secondary infection averted with more frequent screening (efficiency).

## 3. Results

### 3.1. Base Case

#### Cost-Effectiveness

From a societal perspective (Table 2), no screening had a net cost of USD 5000 and was associated with 0.12 QALYs lost per thousand people. Implementing 1×Ag/PCR added USD 11,000 to net costs, and saved 0.035 QALYs per thousand people, resulting in an ICER of USD 320,000 per QALY gained. This was the most cost-effective screening strategy, albeit not attractive by most standards [32,33]. Increasing the testing frequency to 2×Ag/PCR increased costs by USD 9000 and saved an additional 0.016 QALYs per thousand, leading to a greater ICER of USD 561,000 per QALY gained. For both once weekly and twice weekly Ag testing, removing the confirmatory PCR test led to very small incremental health gains but increased net costs due to greater productivity losses. They were thus not cost-effective compared to their PCR-inclusive counterparts, with ICERs over USD 9 million and USD 20 million, respectively. 1×PCR was dominated by 2×Ag/PCR, as it had USD 15,000 additional costs but saved 0.02 fewer QALYs per thousand tested.

From a school perspective (i.e., not considering medical costs and the cost of children’s missed schooling), all strategies involving PCR testing were dominated; they increased costs without any health benefits (Table 3). In this case, 1×Ag was the most cost-effective, with an ICER of USD 324,000 per QALY gained compared to no test. 2×Ag was less cost-effective compared to 1×Ag with an ICER of USD 556,000 per QALY gained.

### 3.2. Unnecessary Days Lost versus Infections Averted

The number of school days unnecessarily missed per secondary infections averted with each testing strategy is displayed in Table 4. Similar to cost-effectiveness, 1×Ag/PCR was the most efficient with 9.5 days needlessly missed per secondary infection averted. When testing frequency was increased to 2×Ag/PCR, the higher rate of false positive test results leads to twice as many days of unnecessary isolation compared to 1×Ag/PCR, but it reduced secondary infections only by 65%. This strategy “cost” over 20 additional days of needless isolation per secondary infection averted compared to 1×Ag/PCR.

1×Ag and 1×PCR were both dominated by 2×Ag/PCR, causing more unnecessary isolation days and averting fewer secondary infections. 2×Ag prevented an additional 0.003 secondary infections per thousand tested over 2×Ag/PCR, however it was an inefficient strategy costing almost 18,000 days of needless isolation per secondary infection averted.

### 3.3. Sensitivity Analyses

Multivariate sensitivity analyses using Monte Carlo simulations showed that findings were robust to uncertainty in input values. In simulations with 5000 iterations, all ICERs were lower than in the base case due to the skewed distributions of key input parameters such as prevalence and medical costs (e.g., 1×Ag/PCR ICER was USD 320,000/QALY in base case as opposed to mean USD 242,000/QALY in sensitivity analyses). Our base-case inputs were closer to zero (bounded) due to the current favorable state of the pandemic in California, but in sensitivity analyses, we assessed higher values such as those that may occur in the content of the surge following the emergence of the delta and omicron SARS-CoV-2 variants.

In these probabilistic simulations, the net cost of 1×Ag/PCR varied between USD 12,000-USD 41,000 per thousand tested, largely due to uncertainty in the prevalence and medical costs, which explained 46% and 27% of variance, respectively. QALYs lost varied between 0.032 to 0.527 per thousand tested with 1×Ag/PCR; the prevalence and QALYs lost per infection accounted for almost 50% and 22% of variance, respectively. The 1×Ag/PCR strategy remained the most cost-effective (yet still unattractive), with a mean ICER of USD 242,000 (95% CI: USD 20,000–USD 953,000) per QALY gained. This outcome was driven primarily by the prevalence of infection (Figure 1), which accounted for nearly 35% of variance. Cost-effectiveness was lower when prevalence of COVID-19 was decreased.

Efficiency outcomes were also improved in sensitivity analyses, similar to cost-effectiveness outcomes. 1×Ag/PCR remained the most efficient across all strategies with a mean efficiency of six needlessly missed school days per secondary infection averted (95% CI: 0.8–22.3). One-way sensitivity analyses showed that this outcome was most influenced by prevalence of COVID-19 and Ag test specificity (Figure 2), which accounted for 38% and 12% of variance, respectively. With this testing strategy, needlessly missed school days varied between 0.8 and 4.3, and between 0.4 and 4.6 secondary infections occurred, per thousand tested.

Threshold analyses revealed that at base-case prevalence (0.2%), the ICER of 1×Ag/PCR vs. no test became favorable (≤USD 100,000/QALY) when R_eff_ reached 2.8. Alternatively, this ICER was obtained when the prevalence of COVID-19 rose to 0.6% at the base-case R_eff_ of 1.1. In two-way analyses where R_eff_ and prevalence were varied simultaneously, the ICER for 1×Ag/PCR vs. no test fell to USD 100,000/QALY gained at R_eff_ = 1.5 and prevalence = 0.4% (similar to California surge in early July 2021). Under these circumstances, doubling the testing frequency to 2×Ag/PCR cost an additional USD 187,000 per additional QALY gained. This value was further reduced when R_eff_ was 2 or greater. In an even worse COVID-19 surge scenario (R_eff_ = 2.0, prevalence = 0.8%, resembling California in October-November 2020), both 1×Ag/PCR and 2×Ag/PCR were highly cost-effective costing USD 19,000 and USD 52,000 per QALY gained, respectively. In this scenario, both testing strategies led to fewer than three needlessly missed school days per secondary infection averted.

Increasing the proportion of recovered individuals (reflecting greater vaccination coverage) exponentially increased the ICER for 1×Ag/PCR compared to no test, with an ICER of USD 600,000/QALY gained when approximately 75% of the population was protected via vaccination or past infection (Figure 3).

During the Omicron surge in California, the prevalence of SARS-CoV-2 infection grew to nearly 6%, [19] while seroprevalence was over 85% providing at least some degree of protection from reinfection [20]. R_eff_ also increased to 1.5 as of 10 January 2022 [21]. Under these conditions, results were significantly altered and 2×Ag testing became the only cost-effective strategy with an ICER of USD 92,800 per QALY gained (compared to 2×Ag/PCR, which had the fewest net costs but led to 0.01 more QALY to be lost per 1000 tested). All other screening strategies were dominated by 2×Ag. However, this option led to over 420 needlessly missed schooldays per infection averted, and thus was not efficient. 1×Ag/PCR was the most efficient strategy leading to 0.23 needlessly missed schooldays per infection averted, followed by 2×Ag/PCR which led to 0.49 needlessly missed schooldays per infection averted.

In further two-way sensitivity analyses, we identified combinations of R_eff_ and prevalence that may affect the choice between no testing, 1×Ag/PCR, and 2×Ag/PCR. For example, when R_eff_ was 1.5 and prevalence was greater than 4%, 2×Ag/PCR was dominant over 1×Ag/PCR. On the contrary, 1×Ag/PCR had potentially prohibitive ICERs when both R_eff_ fell below 1 and prevalence was less than 1% (detail in Appendix A).

## 4. Discussion

We modeled various school-based COVID-19 screening strategies to assess cost-effectiveness and number of unnecessarily missed schooldays to provide evidence for policymakers on asymptomatic screening programs for K-12 schools during the pandemic. Once weekly antigen testing with PCR follow-up for positive tests was the most cost-effective and efficient strategy at all case rates; however, doubling the frequency was also cost-effective and efficient in surge scenarios, such as during the Omicron surge. Circumstances which lead to greater number of COVID-19 cases, such as higher prevalence and transmission rates, or poorer outcomes associated with COVID-19, led to improved cost-effectiveness of 1×Ag/PCR and 2×Ag/PCR testing strategies. Antigen testing alone, regardless of frequency of testing or the level of community transmission, was not cost-effective nor efficient, due to small incremental health benefits and greater numbers of false positive test results compared to antigen testing with PCR confirmation. PCR follow-up allowed false positive antigen test results to quickly be identified and individuals to return to school, eliminating a majority of needlessly missed days and related costs. On the contrary, from a school’s perspective, confirmatory PCR tests increased net costs without apparent health benefits. Providing schools with free or discounted tests and offering operational support should be considered to align incentives.

Our analysis used the effective community transmission value as a proxy for in-school transmission [34] of SARS-CoV-2 and found that at low levels of community transmission such as those in our base case, it might not be cost-effective to conduct regular screening testing at schools. Since in-school transmission is thought to be lower than that in the community when prevention measures are in place, a dynamic approach to testing may be more appropriate where schools pause testing when community transmission is low and resume testing when community transmission is higher (i.e., an outbreak is occurring in the community). Under conditions similar to those during the surge of the Omicron variant in California, twice weekly antigen testing where positive antigen tests are followed by PCR confirmation was a reasonable strategy. This strategy led to the second smallest QALY loss by 0.01 QALYs per 1000 tested and was the second most efficient strategy with less than half a day of needless isolation per infection averted.

Complementary infection mitigation measures (such as face masks, social distancing at school, and vaccination) affect R_eff_ and consequently the cost-effectiveness of school-based testing. Our analysis assumes continued mask-wearing at school, without which, R_eff_ may increase. This suggests that school-based screening may become more cost-effective if mitigation is reduced, but it is not as attractive while mitigation is in place. Vaccination rate also has a similar effect, with cost-effectiveness ratios for school-based screening becoming prohibitive when high proportions of the population is vaccinated. Similarly, testing may permit less rigorous rules and enforcement of infection control measures such as mask wearing and social distancing; however, we did not formally assess this.

Notably, evidence suggests that the incremental benefit of increased testing frequency is significantly altered by the day on which the test is conducted. Screening individuals later in the week lead to decreasing incremental health gains [22,34]. This “weekend effect” occurs because fewer infectious *schooldays* (as opposed to weekends) can be averted when testing towards the end of the week. Our results are in line with this evidence as the twice-weekly (Monday and Thursday) screening strategy led to only a 46% additional benefit over once-weekly (Monday) screening. Our assumption of a Monday testing day for the single weekly test strategy maximized the efficacy of testing.

In addition, there has been fervent debate on how to use antigen tests, based on their imperfect performance [35]. Some criticize their low sensitivity, arguing that the high likelihood of false negatives will give people a false sense of security, thus they should not be used to rule out infection [36,37]. Others argue that while their sensitivity is lower than that of PCR tests, antigen tests can more sensitively detect viral loads that are high enough to permit further transmission, making them more appropriate for screening than PCR, which can detect very low levels of virus and viral debris that are not epidemiologically significant [38,39,40] and lead to unnecessary and burdensome isolation and quarantine. In fact, studies [15,25] have found over 90% antigen test sensitivity when comparison is restricted to PCR positives with lower cycle thresholds (i.e., high viral load, more likely to be transmissible). When we used this higher antigen test sensitivity, antigen testing with confirmatory PCR, both once and twice a week, became slightly more efficient and had a lower cost-effectiveness ratio for moderate transmission. While it is suspected that antigen test sensitivity is reduced for Omicron, the test remains most sensitive to viral loads that are high enough to be transmissible [41]. PCR confirmation for positive tests remained beneficial as long as antigen tests had below perfect specificity. Yet, if antigen tests do have similar sensitivity to PCR tests among asymptomatic individuals, it may be reasonable to repeat the antigen test to confirm positive results, rather than conduct a PCR test, to reduce testing costs and eliminate the need for temporary isolation.

There were several limitations to this analysis. Since this was a cohort model, we were not able to show how increasing the rate of SARS-CoV-2 transmission in school would alter the prevalence (or case rate) in school temporally. These inputs were assessed in sensitivity analyses instead. We were also not able to identify school-specific values for key inputs such as prevalence and transmission rate. Several studies have illustrated that rates of transmission in schools are similar to or lower than those in the community [42], which suggests that our results may be overestimating the cost-effectiveness of school-based testing strategies for communities with low-transmission. On the contrary, our cost-effectiveness findings may be underestimates for smaller schools, which might require hiring additional staff to oversee and implement screening programs. Finally, we did not evaluate results associated with conducting screening on different days of the week. Future studies must consider the impact of alternative testing days, as it is likely that human resource constraints will prevent all schools from following the same screening schedule.

## 5. Conclusions

Public health officials must consider several important tradeoffs when providing testing guidance for the safe operation of schools in California. We compared six asymptomatic screening strategies for COVID-19 and assessed their cost-effectiveness and efficiency to provide an evidence base for designing screening guidance for schools. Our decision model showed that once-weekly antigen testing with confirmatory PCR was the optimal approach for cost-effectiveness and efficiency during moderate transmission periods, yet different pandemic conditions called for different screening strategies to achieve safe environments for in-person learning. Weekly testing was not highly attractive in low community transmission settings and while other infection mitigation measures were in place, suggesting an interplay between different mitigation layers that can be implemented. On the contrary, twice-weekly antigen screening with PCR confirmation was a cost-effective and efficient strategy during widespread community transmission and could be more desirable under surge conditions such as with the Omicron variant. Supporting schools with the financial and human resource costs of such screening programs will be necessary to align incentives under higher transmission scenarios.

## Figures and Tables

**Figure 1 ijerph-19-09371-f001:**
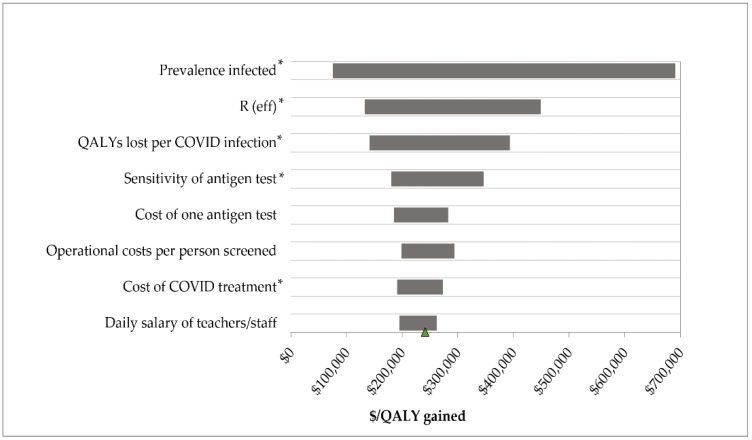
Tornado diagram: one-way sensitivity analyses on the societal incremental cost-effectiveness ratio (ICER) of once weekly antigen testing with confirmatory PCR compared to no test. ICER ranges from USD 76,000 to USD 691,000 per QALY gained. Ag: antigen test, ICER: incremental cost-effectiveness ratio, QALY: quality-adjusted life year, PCR: polymerase chain reaction test, R(eff): effective reproduction number. * Input is inversely associated with ICER.

**Figure 2 ijerph-19-09371-f002:**
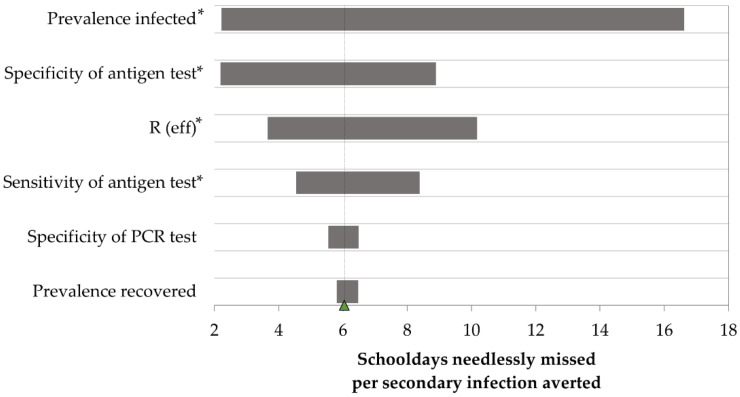
Tornado diagram: One-way sensitivity analyses on the efficiency of once weekly antigen testing with confirmatory PCR compared to no test. Efficiency ranges from 2.2 to 16.6 unnecessarily missed schooldays per secondary infection averted. Ag: antigen test, PCR: polymerase chain reaction test, R(eff): effective reproduction number. * Input is inversely associated with efficiency.

**Figure 3 ijerph-19-09371-f003:**
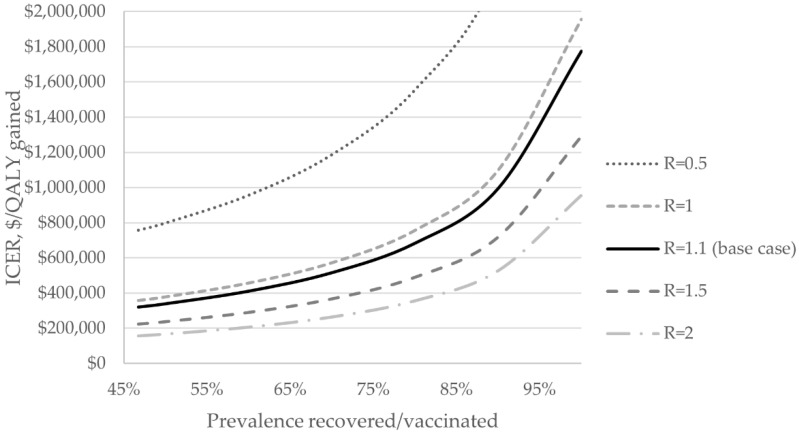
2-way sensitivity analysis on the ICER for once weekly antigen testing with confirmatory PCR compared to no test. Figure capped at USD 2 million per QALY gained. Ag: antigen test, ICER: ncremental cost-effectiveness ratio, PCR: polymerase chain reaction test, R = effective reproduction number.

**Table 1 ijerph-19-09371-t001:** Key input parameter values, ranges, and sources.

Parameter	Base-Case (Range)	Reference
** *Epidemiologic inputs* **		
Prevalence of currently infected *	0.2% (0.05–0.80%)	[19]
Prevalence previously infected or vaccinated (“Recovered”) **	47% (38–55%)	[20]
Effective reproduction number (R_eff_) ***	1.1 (0.5–2)	[21]
QALYs lost per infection	0.078 (0.051–0.206)	[22,23]
** *Test performance inputs* **		
Ag test sensitivity	58% (37–77%)	[15,24]
Ag test specificity	99.6% (99.6–100%)	[24]
PCR test sensitivity	71% (53–83%)	[25,26,27]
PCR test specificity	99% (97–100%)	[25,26,27]
** *Cost inputs* **		
Cost of Ag test	USD 5 (USD 2–USD 8)	Stakeholder input
Cost of PCR test	USD 21 (USD 21–USD 140)	Stakeholder input
Operational costs per person screened (for 1× testing strategies)	USD 4.5 (USD 2.2–USD 6.8)	Stakeholder input
Cost of 1 day of missed schooling (students)	USD 100 (USD 50–USD 200)	Assumed
Cost of 1 day of missed work (teachers/staff, with full productivity loss)	USD 320 (USD 260–USD 480)	[28]

* Percent currently infected is based on California daily case rate as of 13 April 2021, adjusted for a 25% symptomatic rate and 10-day infection period. For Omicron, prevalence is 5.8%, based on daily case rate as of 10 January 2022. ** Prevalence of Recovered is the California antibody prevalence between 28 February–27 March (latest available as of 16 April 2021), this includes both previously infected and vaccinated. For the Omicron analysis, this value is updated to 85.9%. *** R_eff_ for Omicron is 1.5, which is the R_eff_ in California as of 10 January 2022.

**Table 2 ijerph-19-09371-t002:** Cost-effectiveness of testing strategies from a societal perspective, per thousand tested.

Option	Net Cost *	Added Cost *^, ^**	QALYs Lost	QALYs Gained **	ICER (USD/QALYs) **
**No Test**	USD 5264	n/a	0.121786	n/a	n/a
**1× antigen + PCR**	USD 16,439	USD 11,175	0.086904	0.034882	USD 320,358
**1× antigen**	USD 18,824	USD 2385	0.086651	0.000253	USD 9,428,177
**2× antigen + PCR**	USD 25,510	USD 9072	0.070728	0.016176	USD 560,830 ***
**2× antigen**	USD 30,329	USD 4818	0.070490	0.000238	USD 20,243,956
**1×PCR**	USD 40,500	USD 14,990	0.090657	−0.019930	Dominated

Ag, antigen; PCR, polymerase chain reaction; ICER, incremental cost-effectiveness ratio; QALY, quality-adjusted life years. * Costs include weekly cost of testing, productivity loss associated with isolating individuals who test positive for ten days, and medical costs and productivity losses associated with isolating secondary infections when they occur. ** Compared with previous non-dominated strategy. *** Skips 1×Ag which is extended dominated.

**Table 3 ijerph-19-09371-t003:** Cost-effectiveness of testing strategies from the school perspective, per thousand tested.

Option	Net Cost to Schools	Added Cost *	QALYs Lost	QALYs Gained *	ICER (USD/QALYs) *
**No Test**	USD 0	n/a	0.121786	n/a	n/a
**1× antigen**	USD 11,373	USD 11,373	0.086651	0.035135	USD 323,703
**1× antigen + PCR**	USD 11,453	USD 80	0.086904	−0.000253	Dominated
**2× antigen**	USD 20,363	USD 8989	0.070490	0.016161	USD 556,240
**2× antigen + PCR**	USD 20,483	USD 121	0.070728	−0.000238	Dominated
**1×PCR**	USD 28,374	USD 8011	0.090657	−0.020168	Dominated

Ag, antigen; PCR, polymerase chain reaction; ICER, incremental cost-effectiveness ratio; QALY, quality-adjusted life years. * Compared with previous non-dominated strategy.

**Table 4 ijerph-19-09371-t004:** Efficiency of testing strategies, per thousand tested.

Option	Days of Needless Isolation	Added Days of Needless Isolation *	Secondary Infections	Secondary Infections Averted *	Efficiency (Days Needlessly Missed/Secondary Infection Averted) *
**No test**	0.00	n/a	1.571	n/a	n/a
**1×Ag + PCR**	4.3	4.3	1.121	0.450	9.5
**2×Ag + PCR**	8.5	4.3	0.913	0.209	20.4
**1×Ag**	31.9	23.4	1.118	−0.205	Dominated
**2×Ag**	63.7	55.2	0.910	0.003	17,980
**1×PCR**	79.8	71.3	1.170	−0.257	Dominated

Ag, antigen; PCR, polymerase chain reaction. Numbers may not add up due to rounding. * Compared with previous non-dominated strategy.

## Data Availability

The data presented in this study are available in the published article and its Appendix A.

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
