# Peer review of "COVID-19 Testing Strategies for K-12 Schools in California: A Cost-Effectiveness Analysis"

_ijerph, 2022, doi:10.3390/ijerph19159371_

Round 1
Reviewer 1 Report
The topic of this article is very interesting and will also arouse my enthusiasm for COVID-19 testing strategies. Below are a few observations on the study conducted. Some shortcomings are also mentioned in the observations.
1. The conclusion appears insufficient and should connect more closely to the authors research questions that form the basis of the investigation.
2. It is suggested that a literature review, even if only a brief account of information is provided in this form, would prove beneficial to the depth of the study.
Author Response
We thank the reviewer for their encouraging review. Our responses to their two points are below.
Point 1. The conclusion appears insufficient and should connect more closely to the authors research questions that form the basis of the investigation.
Response: Thank you for this observation. We edited the conclusion to reflect the stated goals of the paper more closely (lines 433-440).
Point 2: It is suggested that a literature review, even if only a brief account of information is provided in this form, would prove beneficial to the depth of the study.
Response: We now provide a brief description of the state of asymptomatic screening in schools at the time of the analysis (lines 58-64).
Reviewer 2 Report
With the increasing incidence of COVID-19 and the risk of another pandemic wave in the autumn, it becomes important to define guidelines for screening. This article, outlining the effectiveness of six screening strategies for schools using SARS-CoV-2 antigen and PCR tests with different screening frequencies, allows you to adjust the test model to the level of virus transmission.
The article is methodologically correct. The results of the study are correctly presented. In order to increase the quality of the article, the following corrections are recommended:
(1) modification of the title of the article – the title was formulated too narrowly (the results of the research conducted by the Authors can be successfully used not only in schools in California);
(2) in the Introduction, the results of previous screening tests carried out on university campuses should be presented;
(3) completing the source base of the article (Authors should cite papers that outline the problem of COVID-19 screening in schools, for example, the research report COVID-19 Testing in K-12 Schools: Insights from Early Adopters, as well as articles published in the International Journal of Environmental Research and Public Health);
(4) expanding the catalogue of limitations to the school screening process (in particular those limitations that affect its cost-effectiveness, such as the need for additional staff in smaller schools, or partners for logistics support).
The above comments do not alter the high merit value of the article. Due to the importance of the research problem, which may contribute to the reduction of the number of people admitted to the hospital, it is recommended to continue the study in the future, in particular to investigate the effectiveness of the tests on different days of the week.
Author Response
We thank the reviewer for their enthusiastic response and helpful feedback to improve our manuscript. Our responses to each of their points are below.
Point 1: Modification of the title of the article – the title was formulated too narrowly (the results of the research conducted by the Authors can be successfully used not only in schools in California).
Response: While we agree that the analysis could be applied to other settings in California with some modifications, several of our model parameters were specific to a school population (for example, the proportion of children versus adults and the different probability of severe COVID-19, or the varying societal cost associated with missed schooldays for students versus productivity loss for staff). Therefore, we choose to make the setting explicit in the title.
Point 2: In the Introduction, the results of previous screening tests carried out on university campuses should be presented.
Response: We now include findings from the cited paper (lines 57-58).
Point 3: Completing the source base of the article (Authors should cite papers that outline the problem of COVID-19 screening in schools, for example, the research report COVID-19 Testing in K-12 Schools: Insights from Early Adopters, as well as articles published in the International Journal of Environmental Research and Public Health).
Response: The report “COVID-19 Testing in K-12 Schools: Insights from Early Adopters” is already cited in our Introduction (reference #14). We now elaborate further on the state of school screening strategies at the time of the analysis to strengthen the framing for our research question (lines 59-64).
Point 4: Expanding the catalogue of limitations to the school screening process (in particular those limitations that affect its cost-effectiveness, such as the need for additional staff in smaller schools, or partners for logistics support).
Response: Thank you, we added this important point to the limitations (lines 423-425).
Reviewer 3 Report
Dear, authors
Comgratulations for your interesting contribution focused on the quest for the best form for fighting against the transmission of COVID-19. I think is a clearly publishable paper, but I think the text can improve in two ways:
Despite it is not the aim of the paper, it would be interesting to comment any social indicators or determinants that provokes authorities use another methods for controlling COVIID-19
Also, it would be interesting the authors speak about the different social determinants that can have an effect over the application of that methods depending of clusters as socioeconomic index of families and pupils, sex, level of studies of their parents, and so on
Author Response
We thank the reviewer for their encouraging feedback. While both of the points they have raised are important, these were out of scope for the analysis we have conducted. We had very briefly touched on equitable distribution of resources in our Supplement 2 file; and we now further elaborate on this in the supplement (lines 516-521).